# Host Plants of the Immature Stages of the Invasive Longan Lanternfly, *Pyrops candelaria* (L.) (Hemiptera, Fulgoridae) in Taiwan

**DOI:** 10.3390/insects12111022

**Published:** 2021-11-12

**Authors:** Meng-Hao Hsu, Yueh-Lin Yang, Meng-Ling Wu, Liang-Jong Wang

**Affiliations:** 1Taiwan Forestry Research Institute, Council of Agriculture, Executive Yuan, Taipei City 10079, Taiwan; catflea@tfri.gov.tw (M.-H.H.); mlw@tfri.gov.tw (M.-L.W.); 2Institute of Ecology and Evolutionary Biology, College of Life Science, National Taiwan University, Taipei City 10617, Taiwan; yangyl@ntu.edu.tw

**Keywords:** host plants, invasive species, *Pyrops candelaria*, *Triadica sebifera*

## Abstract

**Simple Summary:**

The longan lanternfly, *Pyrops candelaria* (L.), was recently introduced to Taiwan, and the ongoing spreading of the species may have an impact on native fauna, flora, and environment. The adult of this species is known to prefer the longan tree as a host; however, the preferred host plants for the immature stages have not yet been determined until this study. Thus, we aimed to (1) determine the host range for egg-laying, (2) verify the host plants preferred by nymphs, and (3) analyze the shift of host ranges with different developmental stages. *Triadica sebifera* (L.) Small is the main host plant for nymphs according to our investigations in different locations around northern Taiwan. The results of this study provide great strategic value for decision-makers to allow for effective control of the target tree species.

**Abstract:**

The longan lanternfly, *Pyrops candelaria* (L.), has been invading mainland Taiwan since 2018, but the distribution of the species has been confined to northern Taiwan until now. The manual removal of the adult insects from the longan is still the main control strategy because of the uncertainty around other key host plants, especially for eggs and nymphs. In this study, large numbers of eggs and nymphs were found on *Triadica sebifera* (L.) Small and *Acacia confusa* Merr. The occurrence of immature individuals on *Triadica sebifera* increased with developmental stage from eggs to the last instar from May to July 2021. On 30 April, the first egg mass was recorded. More egg masses were recorded in May, and some could be found in July. In May, only two younger instars were detected. Third and fourth instars began to appear from June, while the fifth instar was mainly recorded from July onwards. The results of this study provide great strategic value for decision-makers to allow for effective control of the target tree species. For now, we proved that longan and pomelo trees, preferred by adults, are not the key hosts for the immature stages of this insect, because few immature individuals were found on them. Therefore, we suggest that the existence of *Triadica sebifera* should be considered when analyzing possible spreading areas of this invasive lanternfly in Taiwan.

## 1. Introduction

The adult longan lanternfly, *Pyrops candelaria* (L.), has long been known for its colorful, spectacular appearance and typical cephalic process. It causes economic losses by sucking the juice from the trunk of fruit trees [1,2,3]. According to Lin et al. [4], the longan lanternfly invaded Taiwan probably from the Kinmen Islands or Hong Kong. This species was first reported in Wugu and Bali in New Taipei City in 2018, and then spread to Tamsui and neighboring Shulin, Beitou in Taipei City [4]. Their analysis of habitat suitability predicted that this invasive species poses a high risk for cultivation areas of pomelo and longan trees. However, whether this lanternfly will spread throughout the plain areas of Taiwan may not solely depend on the distribution of the host plants for adult insects. The existence of the specific host plants for immature stages of the insect may also be important for the establishment of a population in a possible new habitat. Before invading Taiwan, the distribution of the longan lanternfly includes China (Guangxi, Hainan, Guangdong, Fujian, Hong Kong), India, Malaysia, Thailand, Vietnam, Cambodia, Indonesia, and Sikkim [1,5]. Several fruit trees including longan, litchi, olive, orange, and mulberry suffered from its feeding on the trunk and cortex of the trees [1,6]. The host specificity of some insects can change over time, or according to the developmental stage. In the case of the spotted lanternfly, *Lycorma delicatula**,* the host preference changes from a broad range of plant species at the early nymphal stages to a few plant species, such as *Ailanthus altissima,* at the adult stage [7]. Knowledge of the change of host plants can be used to locate the target plant species for the effective control and prevention of spread.

According to Kershaw and Kirkaldy [8], the nymphs of the longan lanternfly feed on *Xanthium strumarium* L., *Urena lobata* L., longan, mango, orange, and pomelo. However, one rarely discovers a nymph in the open, probably due to the cryptic appearance and color. This is the first report to reveal the discovery of many nymphs of this lanternfly on *Triadica sebifera* and other plants. The main goal of this study was to determine which plants are used for laying eggs and which host plants are preferred by nymphs in different developmental stages. The results will help clarify the basic biology and support further studies of the ecological impacts of this invasive species.

## 2. Materials and Methods

### 2.1. Locations

This study was carried out in the locations of Beitou, Shulin, Tamsui and Bali in northern Taiwan, from April to July 2021. In these locations, there were many adult *P. candelaria* found on longan trees [4], so we attempted to find eggs and nymphs in the vicinity. The locations where the search for unhatched egg masses and nymphs was successful are listed in Table 1.

### 2.2. Insect Recording

Investigations have been conducted at least once every week from April to July, and the egg masses and nymphs have not been found until 30 April and 20 May, respectively. The non-living objects or plant species were recorded when individuals in immature stages were observed. Further, the number of unhatched egg masses on each object or plant was also noted down. Nymphs usually congregated and lined the underside of a branch, with intervals in between or on the lower part of a trunk (Figure 1 and Figure 2). Moreover, according to the clear description and illustrations from Kershaw and Kirkaldy (1910) the developing instars can be distinguished into five groups by their distinct morphological traits and sizes, and thus, the number of each instar can be counted. Besides, during the investigation, we carefully identified the species without confusing it with the nymphs of *P. watanabei*, which occasionally mingled with our target insects (Figure 1). The morphological traits of the nymphs of *P. watanabei* are different from those of *P. candelaria*. The detailed description and colorful photographs of each immature stage of the native species were in the project report published by Wang et al. (2011) [9]. Some of the egg masses and nymphs were photographed or collected for recording purposes. A 6 mm Type 1 aspirator (MegaView Science Co. Ltd., Taichung, Taiwan) connected with a vial was used for sucking the smaller, younger three instars; however, for the bigger, fourth and fifth instars, we simply used a 50 mL plastic centrifuge tube to catch it one by one. Egg masses were removed with the attached surface of tree trunks and waited for hatching of the first instars for identification. Occurrences were recorded by visual means, such as via naked eyes or with a Pentax Papilio II 8.5 × 21 telescope (Ricoh imaging Co. Ltd., Tokyo, Japan), and sometimes aided with an MT 14 flashlight (Ledlenser GmbH & Co. KG, Solingen, Germany) in high luminosity (up to 1000 lumens) when necessary, especially on overcast days.

## 3. Results

### 3.1. Egg Masses on Different Plant Species

In total, 49 egg masses were found on 40 objects and plants, including 37 plants, two dead trees, and one wooden rod used for supporting a newly planted *Triadica sebifera* (Table 2). The host plants belonged to 21 species, 14 families. Except for the herbaceous *Alocasia odora*, the rest of the hosts were woody plants. Fifteen (30.6%) egg masses were found on 13 *Acacia confusa*. Three (6.1%) egg masses were recorded on three *T. sebifera*. Only one egg mass was recorded on a longan tree, *Dimocarpus longan*. No egg masses were recorded on a pomelo tree, *Citrus grandis*.

### 3.2. Nymphs on Different Plant Species

In total, 2300 nymphs were found on 248 plants, including 244 trees and four shrubs (Table 3). The host plants belonged to 11 families, including 14 tree species and two shrub species (*Callicarpa formosana* and *Breynia officinalis*). Less than 1% of nymphs were recorded on shrubs, and the nymphs were in the younger three instars. Nearly 80% of nymphs were recorded on *Triadica sebifera*, and 7.1% on *Acacia confusa*. For *T. sebifera,* the maximum of nymphs found on a tree was 133, and the average was 10.7 nymphs per tree. The maximum of nymphs on *Cinnamomum camphora* was 50, due to the occurrence of first instars beside a hatching egg mass. On five species of host plants, including *T. sebifera, Acacia confusa, Lagerstroemia subcostata, Citrus grandis* (pomelo)*,* and *Sapium discolor,* all nymphal stages were recorded. However, only on the former three tree species were more than 100 nymphs (>5%) found. No nymphs were recorded on *Dimocarpus longan*.

### 3.3. The Shift of Host Plant Preference According to the Developmental Stage

To see if the host plant preference shifted, we calculated the percentage of the occurrence of egg masses and each nymphal instar on *Triadica sebifera*. As seen in Figure 3, a trend in the host preference toward *T. sebifera* was increasingly apparent with developmental stage from eggs (6.1%) to the fifth, final nymphal instar (93.1%). Another index in Figure 3 was the number of plant species recorded with >0.5% of egg masses or all nymphal instars, to show whether the range of host plant species was narrow or broad in each developmental stage. A broad range (21) of plant species for egg masses was recorded. However, only a few plant species (nine) were hosts for the younger (first to third) instars, and less than six species were recorded as host plants for the older (fourth to fifth) instars. Hence, the range of host plant species became narrower, with developmental stages from eggs to the last nymphal instar.

### 3.4. Monthly Records of the Immature Stages

As seen in Table 4, in 2021, the first egg mass was recorded in 30 April, and 43 out of 49 egg masses were recorded in May. However, some egg masses were oviposited in July. In terms of nymphs, only the younger two instars were observed in May. In that month, 85.3% and 14.7% of nymphs were in the first and second instar, respectively. The third and fourth instars began to appear from June, while the fifth instar was mainly recorded from July onwards. In June, 12.4%, 30.1%, 42.6%, 14.7%, and 0.2% of recorded nymphs were in the first to fifth instar, respectively. In July, 6.1%, 6.8%, 16.8%, 40.1%, and 30.3% of recorded nymphs were in the first to fifth instar, respectively.

## 4. Discussion

The observations described might indicate the plant species *Triadica sebifera* as the main developing host plant of *Pyrops candelaria*, because many nymphs from the first to fifth instars can be found congregating on the trees. Because *T. sebifera* is also the main host plant for the adults of another lanternfly, *Pyrops watanabei* [9,10], it needs further study to verify whether the invasion of *P. candelaria* would impact on the ecology of the native species. This is the first report indicating that *T. sebifera* is the main host plant for the nymphs of *P. candelaria* according to our investigations in different locations around northern Taiwan. As the insect and the plant both originated from China and occur on the Kinmen Islands, it is worth verifying whether the occurrence of the nymphs on *T. sebifera* is as frequent there as it is in Taiwan.

In this study, we have not found a single nymph of *P. candelaria* on a longan tree, the most preferred host for the adults. Moreover, we frequently observed that the adult lanternflies tend to leave the longan trees after April to move to *T. sebifera* and other plants such as *Sapium discolor* (unpublished data, Hsu), probably for mating and egg-laying. As seen in Table 2 and Table 3, the hosts for immature stages, including eggs and nymphs of the longan lanternfly, are mainly woody plants. The host preference for egg-laying is not as strong as it is for the nymphs, indicating that general site criteria may be more important than species-specific criteria. In the case of *P. candelaria*, this site criterion for oviposition might be the smooth surface of tree trunks or objects (dead trees and wooden rods). In fact, some of the oviposition sites were on the barkless segments of twigs or dead trunks.

The phenomenon that *P. candelaria* occurs more on *T. sebifera* with increasing age of the nymphs might be explained by the theory of defense postulated by Kim et al. (2011) in the case of another lanternfly. They pointed out that through sucking the sap of a host plant with toxic secondary metabolites, spotted lanternflies, *Lycorma delicatula,* may gain some toxic substances in their bodies that they can then use to defend themselves. The evidence of functional morphology in the development of sensory organs on the antenna of *L. delicatula* indicated that the number, size, and complexity of the sensilla increased with nymphal instars, and thus insect behavior such as host plant orientation may be more efficient and specialized in locating the preferred host plants [11]. Fittingly, *T. sebifera* is poisonous to humans and some animals [12,13,14]. Thus, it might explain the host plant changes over time according to the developmental stages of *P. candelaria*.

In this study, the neighboring trees or the trees around the investigated plants were also observed. We found that all plants with immature stages of *P. candelaria* were in the vicinity, probably within flight distance of adults, of longan trees. Moreover, except for *Acacia confusa* and *Sapium discolor*, the plants with nymphs were next to or even in the shade of *T. sebifera*. The nymphs might walk or even directly fall from nearby *T. sebifera* onto other plants. Kim et al. [7] indicated that the nymphs of the spotted lanternfly often fell to the ground because of wind, but then ascended the trees again.

## 5. Conclusions

The results of this study point toward *Triadica sebifera* being the most preferred host plant for the nymphs of *Pyrops candelaria.* Thus, we believe the missing piece of the insect’s life cycle puzzle has been found. We assume that in April adult longan lanternflies begin to leave longan trees because in May mating pairs and egg masses are more likely to be found on a variety of plants near *T. sebifera*. After hatching, the host specificity becomes stronger in later developmental stages; therefore, most of the nymphs tend to move to their nearby preferred host, *T. sebifera.* In June and July, first to fifth instars can be found on *T. sebifera*. Then, several newly emerged adults appear, and it is easy to observe them sucking the sap of the trunk of *T. sebifera.* Later, from August to the end of the year, adult lanternflies tend to leave *T. sebifera* for congregating on longan trees (unpublished data, Hsu). The information in this paper will help predict the movement and ongoing spreading of *P. candelaria* over time, thereby making control measures possible not only for the adults, but also during the immature stages. In fact, during the writing of the manuscript in August, we collected nymphs and adults of *P. candelaria* on *T. sebifera* in Luzhu, Taoyuan City, a new location southwest of the areas reported above. The certainty of host plants will likely be conducive to the successful rearing of immature stages in the laboratory and, thereby, support further research.

## Figures and Tables

**Figure 1 insects-12-01022-f001:**
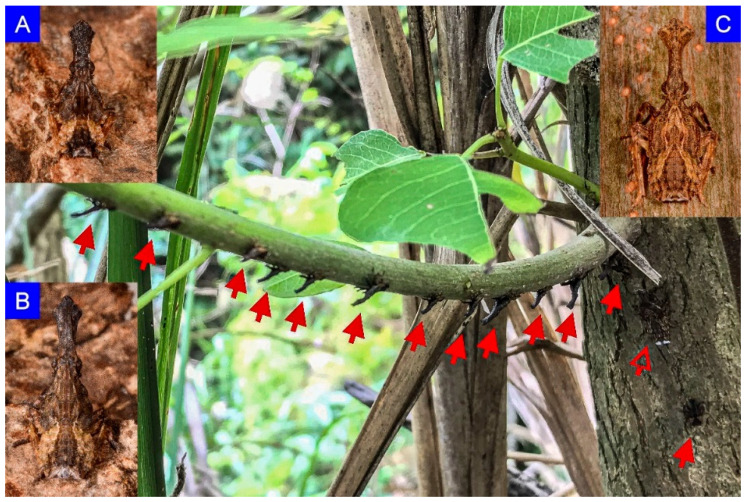
The nymphs of *Pyrops candelaria* usually stay along the underside of the lower branch, with an interval among each other. In this photograph, several second instars (**A**) and third instars (**B**) were found on a branch of *Triadica sebifera*. Each solid arrow points to a nymph. Besides, a fourth instar (**C**) of *Pyrops watanabei*, indicated by a hollow arrow, was observed on the trunk near the base of the green branch.

**Figure 2 insects-12-01022-f002:**
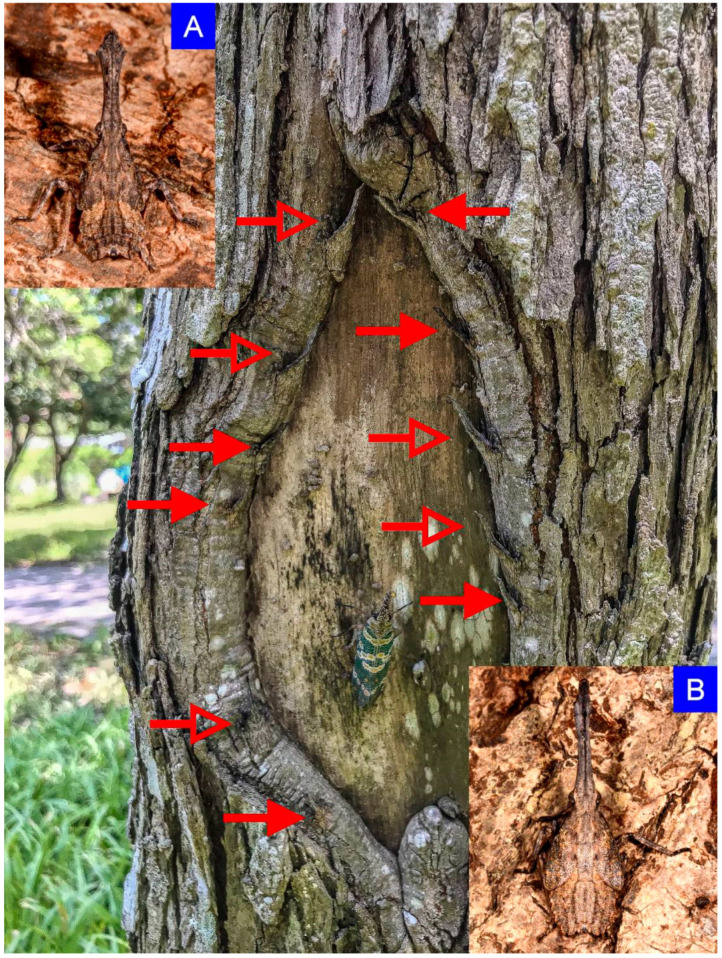
The nymphs of *Pyrops candelaria* may move to the lower part of the trunk in July. In this photograph, six fourth instars (**A**) and five fifth instars (**B**) were circling a newly emerged adult on the trunk of *Triadica sebifera*. The form and color of the nymph are cryptic, and like the remnants of broken twigs and barks. Each arrow points to a nymph. The fourth and fifth instars were indicated by solid and hollow arrows, respectively.

**Figure 3 insects-12-01022-f003:**
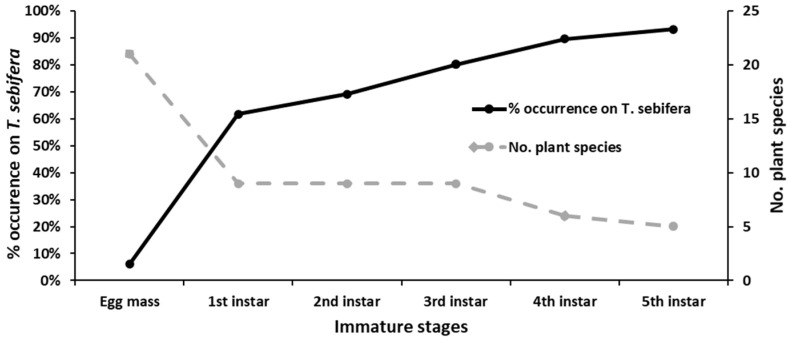
Host plant preference of immature stages from the egg mass to 5th instar of *Pyrops candelaria*. “No. plant species”: The number of plant species with >0.5% of egg masses or nymphs (all insatrs) was recorded.

**Table 1 insects-12-01022-t001:** Description of the locations recorded with the egg masses or nymphs of *Pyrops candelaria* in northern Taiwan, from 30 April to 20 July 2021.

Locations	Description	Major Plant Species	Latitude, Longitude
Guizikeng, Beitou	A park on a hill	Longan, pomelo, and *Triadica sebifera*	25.152, 121.493
Hsing Tian Kong, Beitou Branch	A park of a temple on a hillside	Longan and *Acacia confusa*	25.138, 121.477
Chungho Temple, Beitou	A trail on a hill	Longan and *Sapium discolor*	25.131, 121.508
Donghua Park, Beitou	A park on a hillside	*Acacia confusa*	25.121, 121.508
Zhoumei Village, Beitou	Lowland farm area	Longan and *Triadica sebifera*	25.121, 121.509
Shalaobie Park, Beitou	A park on a hillside	*Triadica sebifera*	25.142, 121.459
Beitou park, Beitou	A park on a hillside, by the road	*Triadica sebifera*	25.137, 121.508
Danfeng Mountain, Beitou	A trail on the mountain at an altitude ca. 120 m	*Sapium discolor*	25.133, 121.513
Grand Hotel, Shulin	A park of the hotel on a hillside	Longan and *Triadica sebifera*	25.081, 121.528
Bali Junior High School, Bali	A graveyard near the school on a hill	Longan and *Triadica sebifera*	25.136, 121.405
Hongshulin Station, Tamsui	A trail by the Metro line on lowland area	Longan and *Triadica sebifera*	25.152, 121.459
Jiantan Park, Shulin	By the road, on the foot of the hill	Longan and *Triadica sebifera*	25.080, 121.525

**Table 2 insects-12-01022-t002:** Number of egg masses of *Pyrops candelaria* recorded on different plant species or objects from 30 April–15 July 2021.

Plant Species or Objects	Family	*n* ^a^	No. Egg Masses
*Acacia confusa*	Fabaceae	13	15
*Ficus subpisocarpa*	Moraceae	2	6
*Celtis sinensis*	Cannabaceae	1	4
*Triadica sebifera*	Euphorbiaceae	3	3
Wood ^b^	-	3	3
*Ficus microcarpa*	Moraceae	2	2
*Ardisia sieboldii*	Primulaceae	1	1
*Koelreuteria henryi*	Spindaceae	1	1
*Alocasia odora*	Araceae	1	1
*Lagerstroemia subcostata*	Lythraceae	1	1
*Semecarpus gigantifolius*	Anacardiaceae	1	1
*Millettia pinnata*	Fabaceae	1	1
*Dalbergia sissoo*	Fabaceae	1	1
*Morus australis*	Moraceae	1	1
*Cordyline fruticosa*	Asparagaceae	1	1
*Dimocarpus longan*	Spindaceae	1	1
*Eucalyptus robusta*	Myrtaceae	1	1
*Machilus zuihoensis*	Lauraceae	1	1
*Machilus thunbergii*	Lauraceae	1	1
*Macaranga tanarius*	Euphorbiaceae	1	1
*Cinnamomum camphora*	Lauraceae	1	1
*Elaeocarpus serratus*	Elaeocarpaceae	1	1
Sum		40	49

^a^ Only the plants or objects with at least a mass of unhatched eggs were counted. ^b^ Two dead trees and a wooden rod were used in the support system for *Triadica sebifera*.

**Table 3 insects-12-01022-t003:** Numbers of nymphs of *Pyrops candelaria* recorded on different plant species from 20 May–20 July 2021.

Plant Species	Family	*n* ^a^	No. Nymphs						%	Mea ^c^	Max ^d^
			1st Instar	2nd Instar	3rd Instar	4th Instar	5th Instar	Total ^b^			
*Triadica sebifera*	Euphorbiaceae	169	229	271	453	535	325	1813	78.8	10.7	133
*Acacia confusa*	Fabaceae	36	5	47	57	40	15	164	7.1	4.2	22
*Lagerstroemia subcostata*	Lythraceae	13	54	35	29	5	1	124	5.4	9.5	24
*Cinnamomum camphora*	Lauraceae	3	50	3	1	0	0	54	2.3	18.0	50
*Psidium guajava*	Myrtaceae	6	5	7	19	11	0	42	1.8	7.0	18
*Citrus grandis*	Rutaceae	6	20	8	10	1	1	40	1.7	6.7	15
*Morus australis*	Moraceae	2	4	7	15	0	0	26	1.1	13.0	21
*Callicarpa formosana*	Lamiaceae	3	3	10	1	0	0	14	0.6	4.7	10
*Sapium discolor*	Euphorbiaceae	3	1	2	2	5	1	11	0.5	3.7	5
*Koelreuteria henryi*	Spindaceae	1	0	0	0	0	3	3	0.1	3.0	3
*Breynia officinalis*	Euphorbiaceae	1	0	1	2	0	0	3	0.1	3.0	3
*Ficus septica*	Moraceae	1	0	0	2	0	0	2	0.1	2.0	2
*Ficus subpisocarpa*	Moraceae	1	0	0	0	0	1	1	0.0	1.0	1
*Diospyros kaki*	Ebenaceae	1	0	1	0	0	0	1	0.0	1.0	1
*Celtis sinensis*	Cannabaceae	1	0	0	0	0	1	1	0.0	1.0	1
*Macaranga tanarius*	Euphorbiaceae	1	0	0	0	0	1	1	0.0	1.0	1
Sum		248	371	392	591	597	349	2300	-	-	-

^a^ Only the plant with at least one nymph was counted. ^b^ Total numbers of the nymphs of all instars. ^c^ Mean numbers of the nymphs found on a plant.^d^ Maximum numbers of the nymphs found on a plant.

**Table 4 insects-12-01022-t004:** Numbers of egg masses and nymphs of *Pyrops candelaria* recorded from April to July 2021.

Month	No. Egg Masses	No. Nymphs					
		1st Instar	2nd Instar	3rd Instar	4th Instar	5th Instar	Total ^a^
April	1	0	0	0	0	0	0
May	43	185	32	0	0	0	217
June	1	116	282	399	138	2	937
July	4	70	78	192	459	347	1146
Sum	49	371	392	591	597	349	2300

^a^ Total numbers of nymphs of all instars.

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
