# Peer review of "Host Plants of the Immature Stages of the Invasive Longan Lanternfly, *Pyrops candelaria* (L.) (Hemiptera, Fulgoridae) in Taiwan"

_insects, 2021, doi:10.3390/insects12111022_

Round 1

Reviewer 1 Report

A nice, well-written and illustrated paper. I’ve only one remark that needs to be documented in the accepted paper: how nymphs of P. candelaria and P. Watanabei were identified and separated is not provided (L.77). It is essential to document how the authors differentiated the different stages between the two species.   The behavior of the species as mentioned looks quite similar to another species of Fulgoridae: Lycorma delicatula. Change and evolution of behavior with older stages in this species have been put in correlation with the development of the sensory function of the sensory organs present on the antenna (Wang et al., 2028). Their function, numbers and their morphology particularly, increase with stages correlatively with a more selective diet spectrum which is observed with this species from poly- to oligo- to even an almost monophagous preference as it seems to be well documented in the present case for P. candelaria. This hypothesis for L. delicatula seems also quite well supported with P. candelaria as well ,and should added and discussed in the discussion and conclusion parts.   Table 3 is the most interesting of the paper and I suggest that the authors add the corresponding figures with curves that will allow to nicely and clearly show how changes between host-plants occur during the different stages of the species (starting with adding the egg masses occurence provided in Table4). The same would be nice for Table 5.   Note that a synthesis about Pyrops candelaria is presented in the FLOW website (see Bourgoin, 2021) together with its distribution and some host-plants not mentioned in the paper (see Constant et al., 2026).   Ref. cited in the review suggested to be added to the paper:   - Bourgoin T. 2021. FLOW (Fulgoromorpha Lists on The Web): a world knowledge base dedicated to Fulgoromorpha.  Version 8, updated [date]. http://hemiptera-databases.org/flow/
- Wang R.R., J.J. Liu, X.Y. Li, A.P. Liang, T. Bourgoin. 2018. Relating antennal sensilla diversity and possible species behaviour in the planthopper pest Lycorma delicatula (Hemiptera: Fulgoromorpha: Fulgoridae). PloS one 13 (3), e0194995

Author Response

Dear Reviewer,

         Thank you very much for your useful comments on our manuscript. In Materials and Methods, we added the sources of literatures for differentiation between P. candelaria and P. watanabei. The description and illustrations or photographs in the above-mentioned two literatures are also helpful for distinguishing the five instars. Actually, via feeding them on branches of Triadica sebifera nymphs of these two species have been reared in the lab from egg masses collected from the field. Besides, we used Mitochondrial genes (COI, ND5, Cyb) to set up the DNA database from the identified adults of above 2 species. We could identify every immature stage (from eggs to 5th instar) accurately by DNA data. So we found that the terga of nymphs of P. watanabei have distinct light patterns that is useful to tell apart from those of P. candelaria. After confirming the morphological difference of every immature stage of these two species, we could recognize every immature stage of these two species in the field investigation.

Thanks for informing us the literature on antennal sensilla in Lycorma delicatula, and this functional morphology evidence is helpful for explanation of insect behavior relating to host-plant orientation. Also, thanks for suggestion of citing FLOW website in this paper to support the description of the distribution and host plants.

Thanks to the comment, we transformed table 4 into Figure 3 to show that the host plant preference shifted according to the developmental stage.

Reviewer 2 Report

I read carefully the submitted mns Insects -1417491,  titled "Host plants of the young stages of the invasive longan lantern -fly, Pyrops candelaria (L.) (Hemiptera, Fulgoridae) in Taiwan". It could be a nice and interesting contribution to the knowledge on  the host plants of the target species that is quite scanty. However, the Introduction section  is lacking for references of previous studies on biology of the target species, P.candelaria and other  co-generic Southeastern  taxa. The AA. do not pointed out the "concept" of host plant : reproducing or feeding plants, or both? In M & M no data or references on morphology or identification methods available have been mentioned. No method of collecting  has been recorded: the AA. speak only about observations and photos/pictures (i.e. the Figures 1 and 2 in the text). These data have to be added in the text because the manuscript  , even as a brief communication, has been submitted to a Q1 Journal (Insects)!. Some periods in the Discussion have to be moved to the Introduction or M & M chapters.  In the Literature List ,the citation 6 (Constant and Pham, 2017) has not been recorded in the text. I suggest to add also the reference  "Constant J., 2015 .The review of the effusus group of the Lanternfly genus Pyrops Spinula 1839" (see the note in the revised mns). Finally, I suggest an English review of the mns by a mother language lecturer.

Author Response

Thank you for your useful comments on our manuscript. In this revised version, we tried our best to add more references on biology of the targeted species or the genus Pyrops. We increased additional 5 literatures, including the description on identification methods of P. candelaria and P. watanabei. The brief description on the collecting of egg masses and some of nymphs also added in “Insect recording”.

Actually, via feeding them on branches of Triadica sebifera nymphs of these two species have been reared in the lab from egg masses collected from the field. Besides, we used Mitochondrial genes (COI, ND5, Cyb) to set up the DNA database from the identified adults of above 2 species. We could identify every immature stage (from eggs to 5th instar) accurately by DNA data. So we found that the terga of nymphs of P. watanabei have distinct light patterns that is useful to tell apart from those of P. candelaria. After confirming the morphological difference of every immature stage of these two species, we could recognize every immature stage of these two species in the field investigation.

The first paragraph of “Discussion” has been omitted, rephrased, and moved to “Introduction.” Thanks for informing us the literatures of Constant J, we listed he and his co-authors’ three references in this revised version. The information of the literatures really enhanced us for the description on the host plants and distribution of Pyrops lanternflies.

As for the concept of host plant…Mostly I would prefer the definition of feeding plants as the host. However, it is easy for oligophagous insects. For most of the insects may have a broad range of feeding plants, and some are the preferred ones which the occurrences of the insects are high. We believe that T. sebifera is the main host plant for the nymph of P. candelaria, and are trying to rear nymphs in the lab. Other plants with the record of the nymph of P. candelaria are not strongly preferred as T. sebifera is. However, we can only say they are not the main host plants. They may still the occasional hosts because many situations will happen as described in this paper “on five species of host plants, including T. sebifera, Acacia confusa, Lagerstroemia subcostata, Citrus grandis (pomelo), and Sapium discolor, all nymphal stages from first to fifth instar were recorded. However, only on the former three tree species more than 100 nymphs (> 5%) were found”.

Reviewer 3 Report

Here the authors report of the potential host plants of immature stages of the invasive species Pyrops candelaria in Taiwan. The experiment of the proposed manuscript is extremely limited in time and scope and is simply preliminary at best. Furthermore, there are a number of significant problems that compromise the value and novelty of the experiment described in this “manuscript” (confusing abstract, extremely poor introduction, missing of essential information in materials and methods, no description of the experimental plan referring to the sampling method, duplicated results etc). Moreover, the English language needs substantial improving. Overall, research is not conducted correctly. Thus, I would not recommend for publication. 

Author Response

Thank you for editing and it is really helpful for us to revise our paper. We did research on P. candelaria and P. watanabei more than two years. At first, it was really difficult to find any egg and nymph of P. candelaria in the field though we carefully searched the longan trees. Luckily, we started to find many egg masses and nymphs on T. sebifera and its neighbouring trees until May because we routinely investigated the seasonal abundance of the adult native P. watanabei. The goal of this brief report is only to report the host plants for nymphs of P. candelaria.  Not only it is the first report for discovery of many nymphs on T. sebifera, but also it provides data enough to show the shift of preference according to developmental stages.

Round 2

Reviewer 1 Report

If possible, closer and bigger photos would be better: fig. 1 can be enlarged to be as wide as the text, and same for the 2 photos of fig.2 (probably only one enlarged of the two presented would be enought).

Author Response

Dear reviewer,

         Thank you very much for your suggestion and we added photographs to present closer looks of nymphs. We hope this supplementary information would better reveal the interesting and cryptic appearance of the nymphs of genus Pyrops.

Reviewer 2 Report

I appreciated the big effort of the Authors to re-draft the submitted manuscript , also considering my previous comments/notes.In particular the adding of more literature references and the nice new Englisgh version. Now,the revised form is clear also in the M&M and Results sections.The Discussion provides appropriate comments to the results obtained , comparing to those recorded by the cited references. The conclusions  are clearly supported by the results. The revised mns (as attached) is suitable to be published. 

Author Response

Dear reviewer,

         Thank you very much for all the comments generously provided and it is helpful for us to enhance the quality of the manuscript.

Reviewer 3 Report

The manuscript (brief note) has now been substantially improved. However, there are still several issues which need to be clarified such as the materials and methods section. Moreover, the language still needs to be improved. Below you will find some of my comments:

Line 76: what is the size of the inspected locations? How many trees or objects did you inspect?

Line 78: The duration of your sampling period is not clear throughout your manuscript. Please state the exact period of your inspections

Line 88: you state that you noted down the number of unhatched egg masses, however there is no such information in the text. Why is the number  of unhatched egg masses so important?

Line 91: replace “follow” with “according to”

Line 138: delete “(Table 3)” from the title and incorporate in the text (end of line 139)

Line 124: delete “(Table 2)” from the title and incorporate in the text (end of line 126)

Line 148: delete “from first to fifth instar”

Line 151: delete “were”

Line 158-159: replace “To see if the host plant preference shifted, two indices were calculated based on Tables 2 and 3. First, the percentage of the occurrence of egg masses and of each nymphal instar on Triadica sebifera were analyzed.” with “To see if the host plant preference shifted, we calculated the percentage of the occurrence of egg masses and each nymphal instar on Triadica sebifera.”

Line 161: replace “age” with “developmental stage”

Line 185: I would recommend to include your weekly sampling in Table 4 instead of monthly in order to show that you found egg masses only in the end of April.

Line 185: Since you found egg masses and nymphs in July why did you not continue your sampling until August? Why did you stop your sampling on July 20? The way you showcase your results are confusing.

Figure 1: This picture is not clear, the two different species cannot be distinguished from this distance. I do not think that the arrows solve this issue.

Figure 2: please see above comment

Author Response

Thank you for editing our manuscript again. Most of the words of this revised version have been changed according to your comments. Thank you for your suggestion, we added photographs to present closer looks of nymphs. We hope this supplementary information would better reveal the interesting and cryptic appearance of the nymphs of genus Pyrops.

Why did us stop our sampling on July 20? We try to put it this way that all immature stages, including eggs and the first to last nymphal instars, have appeared before July 20. And, I assured that there is no sixth instar. (The collected fifth instars of Pyrops candelaria successfully emerged into adults in our lab.) Moreover, the data we collected were enough to show the shift of host plant preference according to developmental stages. We began weekly investigations on immature stages of Pyrops lanternflies from April onwards and luckily find the first egg mass on April 30, 2021. The eggs hatched on May 11 in our lab, so we can confirm the species is P. candelaria. The last egg mass we found was on July 15 (please see title of Table 2). However, we cannot find nymphs until May 20, 2021 (Now, we sure can find nymphs earlier next year!). Therefore, the period presented in this study from May 20 to July 20 (please see title of Table 3) is exactly three months for the record of investigation on nymphs.

Why is the number of unhatched egg masses so important? First, we removed unhatched eggs to our lab and waited for their hatching. The species of first instars can be identified and we tried to rear in the lab. Only the “unhatched egg masses” were counted in the field because it is much difficult to tell the difference between hatched ones laid this year with those of last year. Most of the old, hatched egg masses lost their white, waxy covering, which is important to distinguish between two Pyrops species.

Why not presented our data in Table 4 on a weekly basis? Weekly data are much fluctuated….maybe this week we luckily found many nymphs, but found nothing or less nymphs next week only because the sites or plant species we chose were not so prolific for this insect. We find monthly data better for showing the trend of nymphal stages increasing with time in this study.

Round 3

Reviewer 3 Report

The manuscript (brief note) has now been further improved. The figures are also improved but still remain the weakest point of the ms. I have only a few minor comments:

Line 20: Add "and"  before "(3)"

Lines 20-22: Something is wrong with this sentence.

Line 156: Replace "all nymphs (first to fifth instar)" with "all nymphal instars"

Author Response

Thank you very much for editing our manuscript. We revised the words according to your comments. Especially thank you for suggestion to rewrite the sentence “ Instead of the longan and pomelo, ….”  In this revised version we delete the beginning of the wording, and start directly with “ Tridica sebifera (L.) Small is the main host plant for nymphs according to our investigations in different locations around northern Taiwan.” This sentence will be better for readers to comprehend.